# P3 and NIa-Pro of Turnip Mosaic Virus Are Independent Elicitors of Superinfection Exclusion

**DOI:** 10.3390/v15071459

**Published:** 2023-06-28

**Authors:** Haritha Nunna, Feng Qu, Satyanarayana Tatineni

**Affiliations:** 1Department of Plant Pathology, University of Nebraska-Lincoln, Lincoln, NE 68503, USA; haritha.nunna@huskers.unl.edu; 2Department of Plant Pathology, Ohio Agricultural Research and Development Center, The Ohio State University, Wooster, OH 44691, USA; qu.28@osu.edu; 3United States Department of Agriculture-Agricultural Research Service, University of Nebraska-Lincoln, Lincoln, NE 68503, USA

**Keywords:** Turnip mosaic virus, cross protection, superinfection exclusion, potyvirus, P3, NIa-Pro

## Abstract

Superinfection exclusion (SIE) is an antagonistic interaction between identical or closely related viruses in host cells. Previous studies by us and others led to the hypothesis that SIE was elicited by one or more proteins encoded in the genomes of primary viruses. Here, we tested this hypothesis using Turnip mosaic virus (TuMV), a member of the genus *Potyvirus* of the family *Potyviridae*, with significant economic consequences. To this end, individual TuMV-encoded proteins were transiently expressed in the cells of *Nicotiana benthamiana* leaves, followed by challenging them with a modified TuMV expressing the green fluorescent protein (TuMV-GFP). Three days after TuMV-GFP delivery, these cells were examined for the replication-dependent expression of GFP. Cells expressing TuMV P1, HC-Pro, 6K1, CI, 6K2, NIa-VPg, NIb, or CP proteins permitted an efficient expression of GFP, suggesting that these proteins failed to block the replication of a superinfecting TuMV-GFP. By contrast, *N. benthamiana* cells expressing TuMV P3 or NIa-Pro did not express visible GFP fluorescence, suggesting that both of them could elicit potent SIE against TuMV-GFP. The SIE elicitor activity of P3 and NIa-Pro was further confirmed by their heterologous expression from a different potyvirus, potato virus A (PVA). Plants systemically infected with PVA variants expressing TuMV P3 or NIa-Pro blocked subsequent infection by TuMV-GFP. A +1-frameshift mutation in P3 and NIa-Pro cistrons facilitated superinfection by TuMV-GFP, suggesting that the P3 and NIa-Pro proteins, but not the RNA, are involved in SIE activity. Additionally, deletion mutagenesis identified P3 amino acids 3 to 200 of 352 and NIa-Pro amino acids 3 to 40 and 181 to 242 of 242 as essential for SIE elicitation. Collectively, our study demonstrates that TuMV encodes two spatially separated proteins that act independently to exert SIE on superinfecting TuMV. These results lay the foundation for further mechanistic interrogations of SIE in this virus.

## 1. Introduction

Superinfection exclusion (SIE) or homologous interference is a phenomenon where the primary infection by a virus prevents superinfection by identical or closely related viruses in the same cells and/or host individuals. SIE is a conserved phenomenon among a wide range of plant, animal, and human infecting viruses [1,2]. SIE is also referred to as cross protection when used as a virus disease management strategy in plants to prevent infection by highly virulent strains through purposeful pre-inoculation with a mild isolate of the same virus [3,4]. Cross protection was first used to examine the relatedness of field-collected virus isolates to tobacco mosaic virus (TMV) [5]. Subsequently, tobacco plants pre-infected with mild strains of potato virus X (PVX) were found to be protected from subsequent infection by highly virulent strains of PVX while permitting the infection by unrelated TMV and potato virus Y (PVY) [6,7]. Later, cross protection was found to be effective at controlling several other viruses [8,9]. 

In recent years, virus-encoded proteins have been implicated in SIE activity. For example, the matrix protein of Sonchus yellow net virus (SYNV) [10], p28 of turnip crinkle virus (TCV) [11], NIa-Pro and coat protein (CP) of wheat streak mosaic virus (WSMV) and Triticum mosaic virus (TriMV) [12], p33 and L1L2 of citrus tristeza virus (CTV) [3,13], hemagglutinin encoded by the A56R gene and the serine protease inhibitor SPI-3 (K2) of vaccinia virus [14,15], and the neuraminidase protein of influenza virus [16] have been reported as elicitors of SIE. The molecular mechanisms of SIE remain fully elucidated. RNA-mediated post-transcriptional gene silencing (PTGS) was proposed to be involved in resistance against related superinfecting viruses [17,18,19]. Another proposed mechanism of SIE is that the primary virus interferes with the uncoating of superinfecting viruses through protein-mediated resistance [20,21] or by blocking the crucial proteins required for the replication or movement of the secondary virus [9,22,23]. The third proposed approach is that SIE is a salicylic acid (SA)-mediated defense response activated during the primary virus infection [24,25,26,27]. However, recently it has been proposed that SIE is a virus-encoded function that maintains the fidelity of the viral genome by preventing the replication of progeny viruses in the cells of their own genesis and collaterally targeting highly homologous superinfecting viruses [28,29]. 

Turnip mosaic virus (TuMV), a member of the genus *Potyvirus* in the family *Potyviridae*, is a predominant pathogen of cruciferous crops causing significant yield losses [30]. The genome of TuMV is a single-stranded positive-sense RNA of ~10.0 kilobases, encapsidated in 700 to 750 nm long flexuous filamentous particles. TuMV encodes a single large open reading frame that is translated into a large polyprotein, which was processed into at least 10 mature proteins by the virus-encoded P1 protein [31], the helper-component proteinase (HC-Pro) [32], and the nuclear inclusion body a protein protease (NIa-Pro) [33,34]. Though TuMV is thoroughly studied for viral genes functions, vector transmission, and virus–host interactions [35,36,37,38], TuMV-encoded SIE determinants have yet to be identified. 

In this study, we screened TuMV-encoded cistrons for their potential to elicit SIE against a GFP-tagged TuMV variant in *Nicotiana benthamiana* cells. We found that P3 and NIa-Pro proteins, but not their RNA sequences, efficiently prevented the superinfection of TuMV-GFP. Additionally, we employed the gain-of-function strategy to express TuMV-encoded P3, NIa-Pro, or CP with an RFP-tagged potato virus A (PVA-RFP), a distinct member of the genus *Potyvirus*. We found that *N. benthamiana* leaves systemically infected with the PVA-RFP variant expressing TuMV CP permitted efficient coinfection of TuMV-GFP. By contrast, PVA-RFP-mediated systemic expression of TuMV P3 or NIa-Pro excluded superinfection of TuMV-GFP. Furthermore, regions of P3 and NIa-Pro required for SIE activity were determined through deletion mutation analysis.

## 2. Materials and Methods

### 2.1. Generation of Constructs

#### 2.1.1. TuMV Cistrons in pCASS4

Individual cistrons of TuMV were PCR amplified with cistron-specific primers using pCB-TuMV-GFP [23] as a template. Tobacco etch virus (TEV) leader sequence [39], followed by hemagglutinin (HA) epitope sequence, were fused to the 5′ end of each of TuMV cistron by overlap extension PCR (OL PCR). Gel-isolated OL PCR products were ligated into a binary vector pCASS4 (a variant of pCASS2; [40]) between *Stu*I and *Sac*I restriction sites. A +1-frameshift mutation in the P3 and NIa-Pro cistrons was created by adding an extra nucleotide in oligonucleotides (at nt 27 for P3 and nt 24 for NIa-Pro) by using pCASS-TuMV P3 and pCASS-TuMV NIa-Pro as templates, respectively, for PCR amplification, followed by OL PCR. A series of non-overlapping deletions were introduced into TuMV P3 and NIa-Pro cistrons by OL PCR using pCASS-TuMV P3 and pCASS-TuMV NIa-Pro, respectively, as templates. OL PCR products comprising deletions in P3 or NIa-Pro cistrons were ligated into pCASS4 between *Stu*I and *Sac*I restriction sites. 

#### 2.1.2. pCB-TuMV-RFP

pCB-TuMV-RFP was constructed by replacing the GFP sequence in TuMV-GFP with that of RFP. A DNA fragment between *Stu*I (nt 1127) and *Age*I (nt 2779) restriction sites in TuMV-GFP was amplified by replacing the GFP with that of RFP sequence, followed by ligation into pCB-TuMV-GFP to obtain pCB-TuMV-RFP. 

#### 2.1.3. TuMV Cistrons in PVA-RFP

To express individual cistrons of TuMV in PVA-RFP [41], we first inserted the complete PVA-RFP genome in an intermediate vector pUC57 between *Kpn*I and *Sal*I sites. TuMV P3, NIa-Pro, or CP cistrons were inserted into PVA-RFP between NIb and CP cistrons using *Mlu*I and *Sac*II restriction sites. These mutants were cloned into pRD400 binary vector between *Sal*I and *Kpn*I restriction sites.

The presence of inserts and introduced mutations in plasmid DNAs was confirmed by nucleotide sequencing at Genewiz, Inc. (Burlington, MA, USA). 

### 2.2. Superinfection Exclusion Assay

The binary plasmids with individual TuMV cistrons, TuMV-GFP, PVA-RFP, PVA-RFP with TuMV P3, NIa-Pro, or CP, and WSMV P1 were transformed into *A. tumefaciens* strain C58C1 using the chemical transformation method. The *Agrobacterium* culture was grown in Luria-Bertani (LB) liquid medium at 28 °C overnight, and the agrobacteria pellets were suspended in 10 mM MES, pH 5.5 containing 10 mM MgCl_2_ and 100 µM acetosyringone to an optical density of 0.5 at 600 nm. The agrosuspensions were incubated at room temperature for 3–4 h. The agrocultures with individual TuMV cistrons (except HC-Pro) in binary plasmids were equally mixed with a silencing suppressor WSMV P1 and infiltrated into *N. benthamiana* on the abaxial side of the leaf and incubated at 24 °C in a growth chamber with a 16 h photoperiod. *N. benthamiana* leaves agroinfiltrated with individual cistrons of TuMV were super infiltrated with agrosuspension harboring pCB-TuMV-GFP at 24 h post-agroinfiltration. At 3 days post-challenge agroinfiltration (dpcai), the agroinfiltrated leaves were examined under a confocal microscope. The *Agrobacterium* cultures harboring pRD400-PVA-RFP-TuMV cistrons were mixed with an equal volume of pPZP-WSMV P1 and infiltrated into *N. benthamiana* leaves. At 10 days post-agroinfiltration (dpai), the symptomatic top leaves were challenge-agroinfiltrated with pCB-TuMV-GFP. The upper non-infiltrated leaves were collected at 10 dpcai and observed under a confocal microscope. At least two independent clones per construct were used for SIE assay and results presented from one representative clone per construct. SIE experiments were repeated at least 3–4 times, and results from one representative experiment were presented. 

### 2.3. Confocal Microscopy

The presence of GFP or RFP in the epidermal cells of *N. benthamiana* leaves at 3 dpcai were examined with a Nikon 90i upright fluorescent confocal microscope using GFP (at 488 nm excitation and 500 to 550 nm emission) and RFP (at 561.4 nm excitation and 570 to 620 nm emission) filters. 

### 2.4. Western Blot

Agroinfiltrated *N. benthamiana* leaves with individual cistrons of TuMV were collected at 72 h post-agroinfiltration in mesh bags (Agdia, Elkhart, IN, USA) for total protein isolation. Total proteins were extracted by grinding the leaves in protein extraction buffer [50 mM Tris-acetate pH 7.4, 10 mM Potassium acetate, 1 mM EDTA, 5 mM DTT containing 1 Complete Mini Protease Inhibitor Cocktail tablet (Roche, Indianapolis, IN, USA)] and mixing the extract with an equal volume of 2X Laemmli buffer [42]. The protein extract was incubated in boiled water for 3 min, followed by centrifugation at 16,000× *g* for 3 min. The supernatant containing the total protein was incubated at −20 °C. Fifteen microliters of total proteins was separated through 4 to 20% gradient Tris-Glycine-SDS polyacrylamide gels (Invitrogen, Carlsbad, CA, USA), followed by transfer onto PVDF membranes using iBlot dry blotting system (Invitrogen). The PVDF membrane was incubated in a blocking solution of 5% (*w*/*v*) nonfat dry milk powder for 1 h, followed by anti-HA monoclonal antibody or GFP-specific monoclonal antibody at 1:10,000. Anti-mouse HRP conjugate at 1:50,000 was used as the secondary antibody. The blot was developed by incubating for 5 min in a 1:1 mixture of HRP substrate peroxide solution and HRP substrate luminol reagent (Millipore, Billerica, MA, USA). The Molecular Imager Chemi-Doc XRS with Image lab software system was used for the development of immunoreactive protein bands on PVDF membrane. The large subunit of RuBisCo protein separated through 4 to 20% SDS-PAGE was included as a control for the amount of protein loaded per well in Western blots. 

## 3. Results

### 3.1. TuMV Exhibits SIE

We examined whether TuMV exhibited SIE by inoculating *N. benthamiana* plants with GFP- and RFP-tagged variants of TuMV (TuMV-GFP and TuMV-RFP) (Figure 1A). *Agrobacterium* harboring pCB-TuMV-GFP and pCB-TuMV-RFP were co-infiltrated along with pPZP-WSMV P1 [43], a suppressor of RNA silencing, into *N. benthamiana* leaves. At 10 dpai, the upper non-infiltrated leaves were observed under a confocal microscope for the expression of GFP and RFP, and it was found that GFP and RFP were rarely co-expressed in the same cells (Figure 1B). Rather, GFP- or RFP-tagged TuMV variants were found to occupy islands of cells in a mutually exclusive pattern, with very few cells expressing both GFP and RFP at the boundaries of cell islands (Figure 1B, merged). These data suggest that *N. benthamiana* cells infected by TuMV-GFP or TuMV-RFP prevented the replication of the sister variant in a mutually exclusive manner, indicating that TuMV exhibited SIE in cells of *N. benthamiana*.

### 3.2. Screening TuMV-Encoded Cistrons for SIE Activity

The above experiments revealed that TuMV exhibited SIE in *N. benthamiana* cells. We next screened the individual cistrons encoded in the TuMV genome for potential determinants of SIE. To this end, the coding sequence of each of the TuMV proteins fused with an N-terminal HA epitope was inserted into the binary vector pCASS4. *Agrobacterium* suspensions harboring the resulting recombinant constructs were co-infiltrated with pPZP-WSMV P1 into *N. benthamiana* leaves. At 3 dpai, total proteins isolated from these leaves were examined for the expression of corresponding proteins with Western immunoblot using an anti-HA antibody. Except for P1 (low expression) and 6K1 (undetectable), all other TuMV proteins accumulated to easily detectable levels (Figure 2A).

We next examined the transiently expressed TuMV proteins for their potential to elicit SIE against TuMV-GFP. To this end, the protein-expressing constructs were delivered into *N. benthamiana* leaves first, followed by the delivery of pCB-TuMV-GFP 24 h later. As shown in Figure 2B, *N. benthamiana* leaf cells pre-expressing TuMV P1, HC-Pro, 6K1, CI, 6K2, NIa-VPg, NIb, or CP permitted the efficient expression of GFP from the replicating TuMV-GFP. By contrast, leaf cells pre-expressing TuMV P3 or NIa-Pro blocked the expression of GFP to near completion (Figure 2B). These results were further verified with Western blotting using a GFP antibody (Figure 2C). Together, our data suggested that TuMV P3 and NIa-Pro independently elicited SIE against the superinfecting TuMV-GFP. 

### 3.3. The P3 and NIa-Pro Proteins, Rather than Their Coding RNA, Are Responsible for SIE Elicitation

We established in the previous section that P3 and NIa-Pro cistrons act independently to block the replication of TuMV-GFP. However, the possibility exists that this effect was caused by the P3 and NIa-Pro-encoding RNA being exceptionally potent templates for the biogenesis of siRNAs; the latter could then target the genome of TuMV-GFP for degradation through RNA silencing. To assess this possibility, single nucleotide insertions were introduced into the P3 and NIa-Pro reading frames at the ninth and eighth codons, resulting in a stop codon after the fourteenth and ninth amino acid, respectively (Figure 3A). The P3 and NIa-Pro cistrons with +1 frameshift mutation were ligated into pCASS4 between 35S and terminator sequences. As shown in Figure 3B, the frameshifted P3 and NIa-Pro mutants, upon transient pre-expression through agroinfiltration, completely abolished the SIE-eliciting activities of the wildtype P3 and NIa-Pro. Therefore, SIE elicitation by P3 and NIa-Pro was mediated by the encoded protein rather than their mRNAs. 

### 3.4. TuMV-GFP and PVA-RFP Readily Co-Infect the Same Cells

To further corroborate the SIE-eliciting capacity of TuMV P3 and NIa-Pro in a different system, we next attempted to express these proteins using PVA, a virus distantly related to TuMV. Both TuMV and PVA are members of the genus *Potyvirus*, with a 54% polyprotein identity. To first determine whether these two viruses exert SIE on each other, *Agrobacterium* suspensions harboring pRD400-PVA-RFP and pCB-TuMV-GFP (Figure 4A) were co-agroinfiltrated into *N. benthamiana* leaves. At 10 dpai, the upper non-infiltrated leaves were examined under a confocal microscope. We found that GFP and RFP, resulting from the replication of TuMV-GFP and PVA-RFP, respectively, coexisted in a large number of cells (Figure 4B). Thus, TuMV and PVA readily infected the same cells, and thus they were not mutually exclusive.

We next examined whether PVA could superinfect *N. benthamiana* leaves systemically infected by TuMV-GFP by first introducing pCB-TuMV-GFP, followed by pRD400-PVA-RFP introduction at 10 dpai. At 10 days after PVA-RFP introduction, the upper non-agroinfiltrated leaves were examined under a confocal microscope. PVA-RFP was able to infect cells pre-infected by TuMV-GFP, as both RFP and GFP were found in large numbers of cells. Indeed, ~85% of *N. benthamiana* cells expressed both RFP and GFP (Figure 4C).

These data suggest that TuMV and PVA did not exhibit mutual SIE to each other in *N. benthamiana* cells. As a result, PVA-RFP could be used to express the heterologous TuMV proteins.

### 3.5. PVA Encoding TuMV P3 or NIa-Pro, but Not CP, Exerted SIE to TuMV-GFP

Since TuMV and PVA co-infected a large proportion of cells of *N. benthamiana*, we next utilized PVA as an expression vector for the transient expression of TuMV P3, NIa-Pro, or CP, followed by challenge inoculation with TuMV-GFP. TuMV CP was utilized as a negative control. The P3, NIa Pro, or CP cistrons were first inserted into PVA-RFP in an intermediate vector pUC57, then transferred into a binary vector pRD400. The N- and C-termini of the three TuMV proteins were fused with 26 N-terminal amino acids of PVA CP and 18 amino acids of PVA NIb, respectively, to ensure the efficient cleavage of the protein products [44] (Figure 5A). As a result, TuMV-encoded proteins will have 26 N-terminal amino acids of PVA CP at the N-terminus, and 18 C-terminal amino acids of PVA NIb sequence at the C-terminus of the inserted proteins of TuMV (Figure 5A).

*Agrobacterium*-harboring pRD400-PVA-RFP with TuMV proteins were infiltrated into *N. benthamiana* leaves along with a silencing suppressor pPZP-WSMV P1 at a 1:1 ratio. At 10 dpai, the symptomatic upper non-inoculated leaves were challenge-infiltrated with an agrosuspension carrying pCB-TuMV-GFP. At 10 dpaci, systemic leaves pre-infected with PVA-RFP expressing TuMV CP permitted the superinfection of TuMV-GFP (Figure 5B, bottom panel). Cells glowing yellow represented those replicating both PVA-RFP-TuMV CP and TuMV-GFP. In contrast, systemic leaves infected with PVA-RFP encoding TuMV P3 or NIa-Pro revealed the expression of both GFP and RFP, but in different areas of the leaves without the visible co-infection of cells by PVA-RFP and TuMV-GFP (Figure 5B, top and middle panels). These results suggest that there was a mutual exclusion between PVA-RFP-TuMV P3 or NIa-Pro and TuMV-GFP in a substantial proportion of cells that would have been expected to have received both replicons. These data further confirmed that TuMV P3 and NIa-Pro acted as independent SIE elicitors of TuMV. We found a few cells of *N. benthamiana* that were pre-infected by PVA-RFP harboring TuMV P3 or NIa-Pro were co-infected with TuMV-GFP. This might be due to insert instability beyond 10 dpi and/or the low-level expression of TuMV proteins through PVA-RFP. 

### 3.6. Mapping the Minimal Regions of TuMV P3 and NIa-Pro Required for SIE Activity

We next attempted to delineate the minimal regions of TuMV P3 and NIa-Pro required for SIE elicitation. This was undertaken by introducing a series of nonoverlapping deletions with HA-TuMV P3 and HA-TuMV NIa-Pro. In P3, nine deletions encompassing amino acid residues 3 to 40, 41 to 80, 81 to 120, 121 to 160, 161 to 200, 201 to 240, 241 to 280, 281 to 320, and 321 to 352 were generated (Figure 6A). Similarly, six deletions were introduced into the NIa-Pro, encompassing amino acid residues 3 to 40, 41 to 80, 81 to 120, 121 to 160, 161 to 200, and 201 to 242 (Figure 7A).

We found that amino acid residues 241 to 352 of P3 could be deleted without dramatically compromising the SIE of TuMV-GFP, whereas amino acid residues 3 to 200 appeared to be indispensable for its SIE-eliciting activity (Figure 6B). Thus, amino acids 3 to 240 of P3 are in the minimal region required for SIE elicitor activity. By contrast, only one deletion mutant of NIa-Pro (deleting amino acid residues 41 to 80) retained the SIE-eliciting activity (Figure 7B). Therefore, two distinct domains of NIa-Pro comprising amino acid residues 3 to 40 and 81 to 242 likely functioned cooperatively to exert SIE on TuMV-GFP. 

## 4. Discussion

Recent research efforts by multiple groups strongly suggest that SIE reflects a virus-encoded functionality [28,29]. As a result, identifying virus-encoded SIE elicitors is expected to foster a mechanistic understanding of SIE, and facilitate the development of management strategies targeting SIE elicitors. In this study, we have investigated SIE elicitors encoded by the TuMV genome through the transient overexpression of each of the TuMV cistrons except for P3-PIPO, followed by superinfection with TuMV-GFP. The candidate SIE elicitors were subsequently verified by their heterologous expression through a distantly related virus (PVA-RFP). A similar gain-of-function strategy was used to determine the SIE effectors of WSMV by engineering individual cistrons into TriMV, and vice versa [12]. We found that the expression of TuMV P3 or NIa-Pro (but not that of other TuMV proteins) blocked the superinfection of TuMV-GFP. The further deletion mutagenesis of TuMV P3 and NIa-Pro coding regions revealed that amino acids 3 to 240 of P3 and 3 to 40 and 81 to 242 of NIa-Pro are required for their SIE-eliciting activity. 

Importantly, the SIE elicitation by P3 and NIa-Pro was unlikely to have been caused by RNA silencing induced through their coding RNA. This was demonstrated by shifting their reading frames with the insertion of a single nucleotide early in their respective ORFs. Such minor insertions were expected to cause minimal perturbation in the sequences of the coding RNA, yet completely abolished SIE activity against TuMV-GFP. Interestingly, we found that either of the two proteins was sufficient to elicit SIE against TuMV, though the reason for having more than one SIE elicitor remains to be investigated. It is possible that TuMV utilizes two distinct mechanisms to elicit SIE. Several viruses have been reported to encode a single SIE elicitor in their genomes. For example, SYNV, TMV, and TCV encode the matrix protein, CP, and p28, respectively, to elicit SIE activity [10,11,20]. In contrast, only a few other viruses, like CTV, WSMV, and TriMV, have been reported to encode more than one protein as SIE effectors [12]. The identification of P3 and NIa-Pro of TuMV [this study], CP and NIa-Pro of WSMV and TriMV [12] as SIE elicitors revealed that the effectors of SIE are not completely conserved at the family level, as all these three viruses belong to the family *Potyviridae*. 

The mechanism of SIE is not clearly known, as different groups postulated different mechanisms. In SIE, the replication of the superinfecting virus is one of the main targets in plant and animal viruses. In TCV, p28, a replication protein of the primary infecting virus, targets the same protein encoded by a superinfecting virus [28]. Recently, Perdoncini Carvalho et al. (2022) suggested that SIE is the manifestation of Bottleneck, Isolate, Amplify, and Select (BIAS), a model proposed to explain the persistent proliferation of the beneficial mutations and relentless purging of lethal errors through natural selection imposed on viral genome copies that become isolated in separate cells by virus-endowed bottlenecking. On the other hand, in Sindbis virus and Semliki Forest virus infections, a virus-encoded proteinase was found to hinder the replication of superinfecting viruses by incorrectly processing the proteins involved in replication [45,46]. Consistent with the potential role of virus-encoded proteases, the NIa-Pro proteases encoded by TriMV and WSMV, two distantly related potyviruses, have now been identified as SIE elicitors [12]. Separately, a protease encoded by CTV [13] was also found to elicit SIE. 

Additional possible mechanisms of SIE include interfering with the virion assembly and/or disassembly of challenging viruses [47,48]. Challenge inoculation of TMV-infected plants with either naked RNA or TMV strains coated with brome mosaic virus CP successfully infected the cells, demonstrating that the CP of TMV likely acts on the superinfecting virus by preventing the uncoating of the latter [47]. Similarly, the CP of PVA was found to act at the initial stages of infection by preventing the re-encapsidation of superinfecting virus virions [48]. Finally, RNA silencing, a sequence-specific RNA degradation mechanism of the host plants, was also proposed as one of the possible mechanisms of SIE that target the RNA of superinfecting viruses [49,50]. However, as discussed earlier, RNA silencing does not explain SIE elicitation by TuMV P3 and NIa-Pro. 

The deletion analysis of P3 and NIa-Pro revealed that deleting amino acids 3 to 220 of P3, or 3 to 40 and 81 to 242 of NIa-Pro, abolished their SIE-eliciting activity. The C-terminal region of TuMV P3 has a hydrophobic domain, the deletion of which caused the virus to be replication-defective [51], whereas the N-terminal region covered by our deletion (amino acids 3 to 220) did not appear to be critical for replication. Thus, it appears that two different regions of P3 confer distinct functionality during virus infection, with an N-terminal region conferring SIE and a C-terminal region supporting replication. The successful identification of two TuMV-encoded SIE elicitors would lay the foundation for further mechanistic investigations and the development of SIE-targeting virus control strategies. The SIE elicitors would help to examine how the elicitors would prevent the co-infection of incoming closely related viruses, and such mechanisms would define novel management strategies against virus diseases of plants, animals, and humans. 

## Figures and Tables

**Figure 1 viruses-15-01459-f001:**
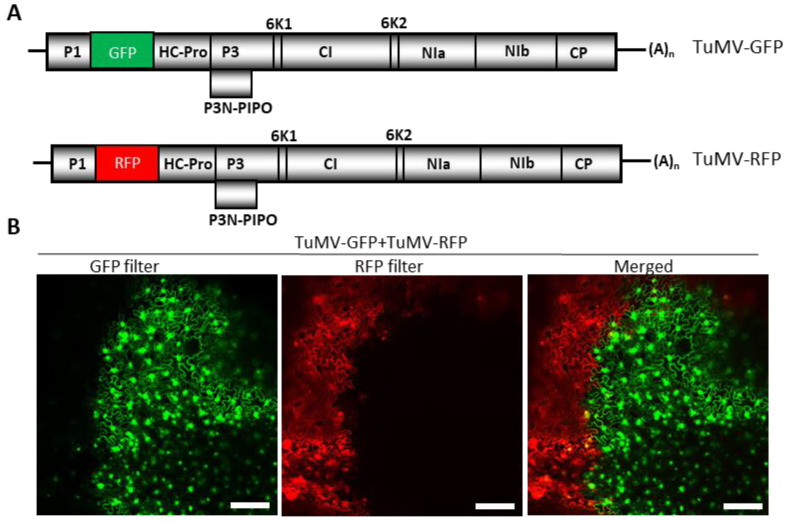
Demonstration of superinfection exclusion of Turnip mosaic virus (TuMV). (**A**) Schematic representation of the genomes of two variants of TuMV with GFP (TuMV-GFP) or RFP (TuMV-RFP) genes inserted between P1 and HC-Pro cistrons. (**B**) *Nicotiana benthamiana* leaves were co-agroinfiltrated with TuMV-GFP and TuMV-RFP, followed by an examination of the expression of GFP and RFP under a confocal microscope. The ‘merged’ image is a superimposed image showing the expression of GFP and RFP, depicting the mutual exclusion of two variants of TuMV. Bars:100 µm.

**Figure 2 viruses-15-01459-f002:**
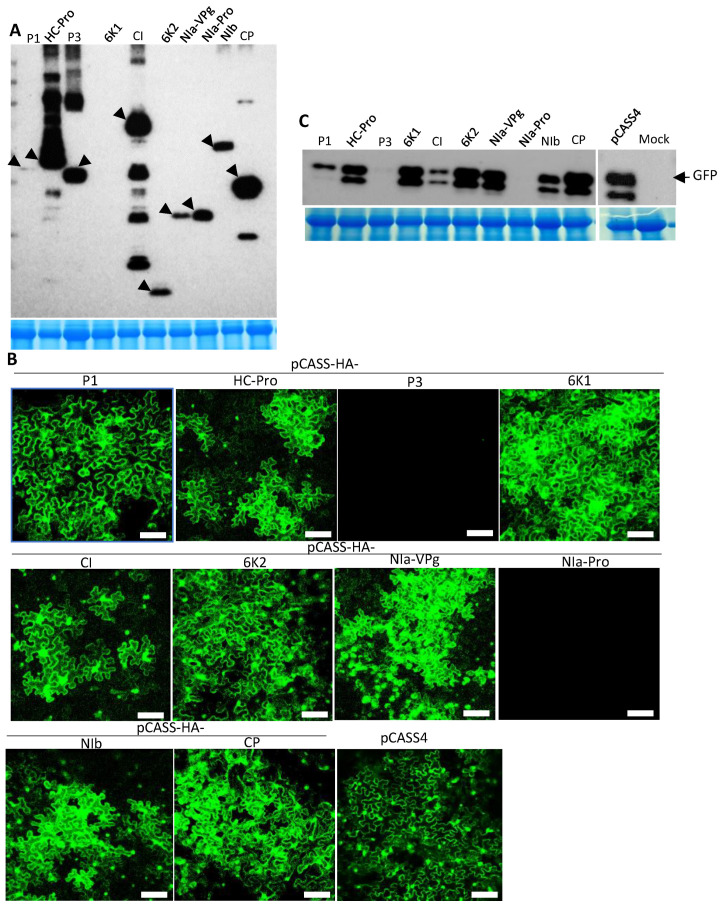
Screening of the Turnip mosaic virus (TuMV) genome for elicitors of superinfection exclusion. TuMV-encoded cistrons were engineered into pCASS4 between 35S promoter and terminator sequences, followed by agroinfiltration into *Nicotiana benthamiana* leaves. The agroinfiltrated leaves were challenge agroinfiltrated with pCB-TuMV-GFP at 24 h post-agroinfiltration. (**A**) Western blot of total proteins extracted from agroinfiltrated HA-tagged TuMV cistrons with anti-HA antiserum. Arrowheads indicate expected TuMV proteins. The Coomassie-stained SDS-PAGE shows the RUBISCO protein for the amount of protein loaded per well. (**B**) Expression of GFP in pre-agroinfiltrated *N. benthamiana* leaves with TuMV cistrons that were challenge-agroinfiltrated with pCB-TuMV-GFP at 3 days post-agroinfiltration. Pre-agroinfiltrated TuMV cistrons were indicated on top of each picture. All TuMV cistrons except P3 and NIa-Pro facilitated the expression of GFP. Bars: 100 µm. (**C**) Western blot analysis of total proteins from pre-agroinfiltrated *N. benthamiana* leaves that were challenge-agroinfiltrated with pCB-TuMV-GFP. The blots were treated with anti-GFP monoclonal antiserum. The Coomassie-stained gel expressing the RUBISCO protein for the amount of protein loaded per well.

**Figure 3 viruses-15-01459-f003:**
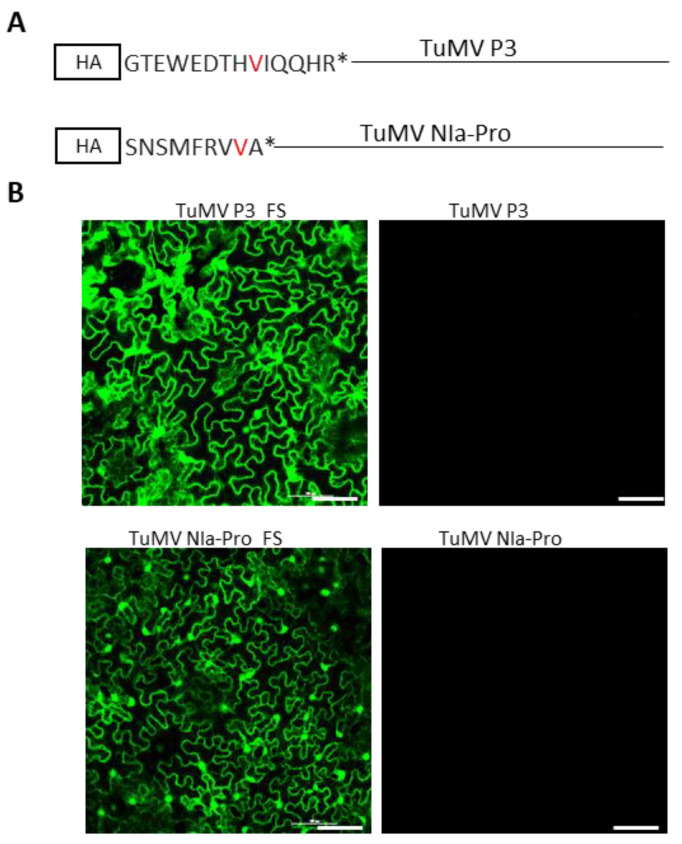
Turnip mosaic virus P3- and NIa-Pro-encoded proteins, but not the RNA sequences, are required for SIE activity. (**A**) Schematic representation of +1 frameshift mutations (red-colored amino acids represent the change in amino acid due to a +1 frameshift) into HA-tagged TuMV P3 and NIa-Pro cistrons. The +1 frameshift mutations resulted in a stop codon (asterisk) after 14 and 9 amino acid codons in P3 and NIa-Pro cistrons, respectively. (**B**) Expression of GFP in challenge agroinfiltrated with TuMV-GFP in *Nicotiana benthamiana* leaves that were pre-agroinfiltrated with agrosuspension harboring TuMV-P3 + 1FS, TuMV-P3, TuMV-NIa-Pro + 1FS, or TuMV-NIa-Pro at 3 days post-challenge agroinfiltration. Bars: 100 µm.

**Figure 4 viruses-15-01459-f004:**
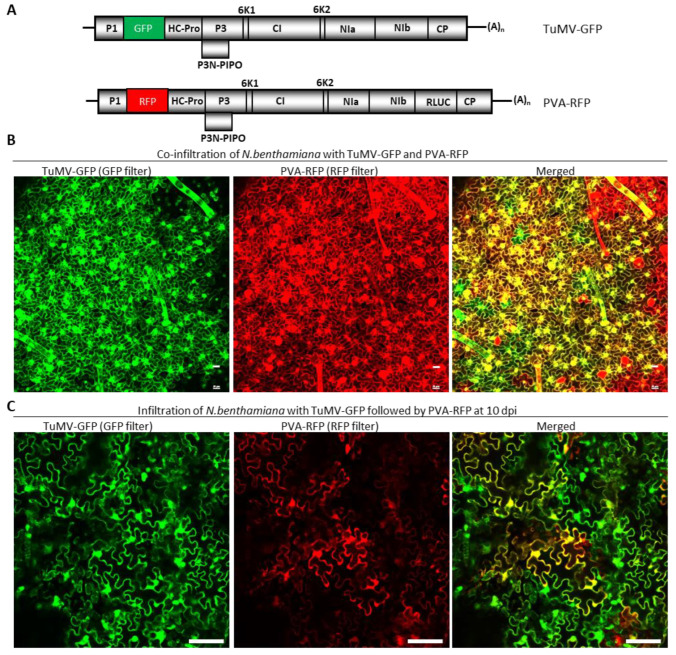
Potato virus A and Turnip mosaic virus can co-infect the same cell. (**A**) Schematic diagrams showing the genomic organizations of TuMV-GFP and PVA-RFP. (**B**) Expression of GFP and RFP in *Nicotiana benthamiana* leaves co-agroinfiltrated with pRD400-PVA-RFP and pCB-TuMV-GFP. The ‘merged’ image on the right is a superimposed image expressing both GFP and RFP. The yellow-colored cells indicate the co-infection of both viruses. (**C**) Expression of GFP and RFP in *N. benthamiana* leaves that were first agroinfiltrated with pCB-TuMV-GFP, followed by super agroinfiltrated with pRD400-PVA-RFP. The upper non-agroinfiltrated *N. benthamiana* leaves were observed in a confocal microscope for the expression of RFP and GFP resulting from the replication of PVA and TuMV, respectively. The merged image on the right shows the co-infection of TuMV-GFP and PVA-RFP. Bars in B and C are 20 µm and 100 µm, respectively.

**Figure 5 viruses-15-01459-f005:**
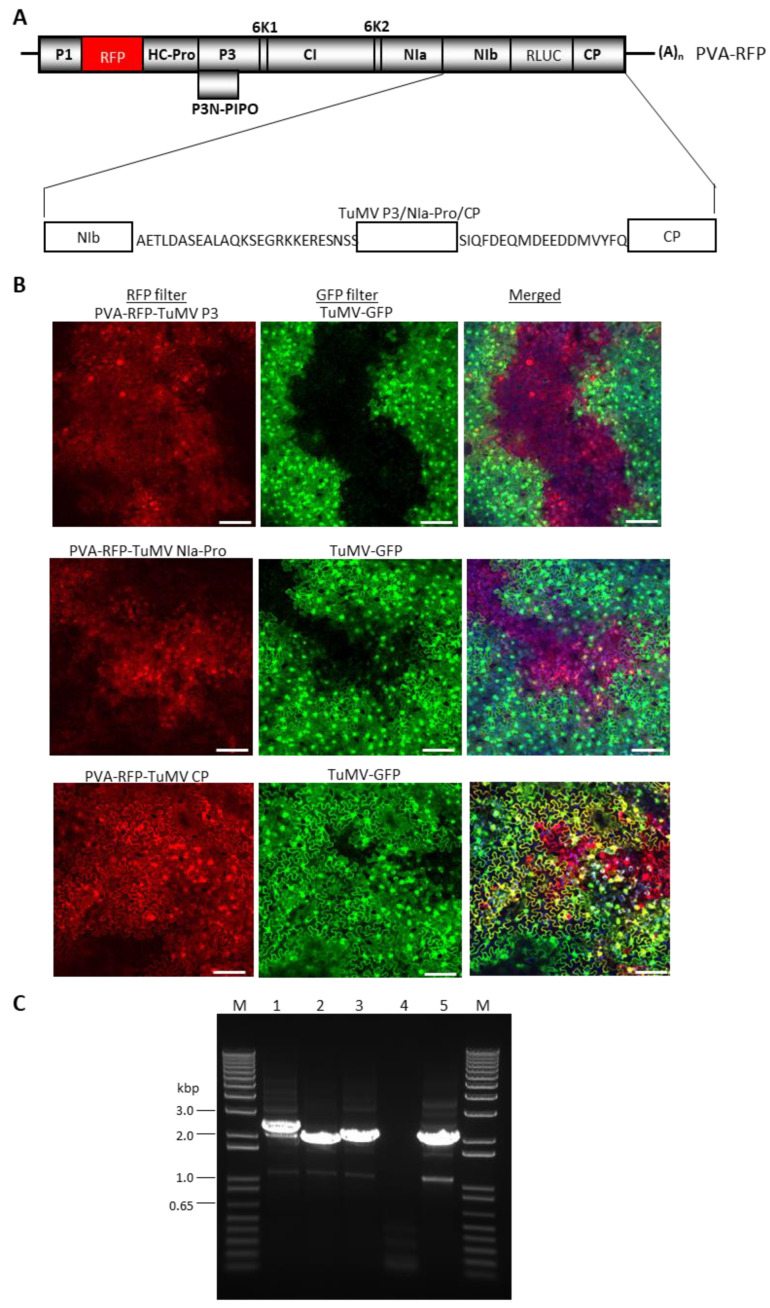
Superinfection exclusion assay of select Turnip mosaic virus cistrons using the add-a-gene strategy in potato virus A (PVA). (**A**) Schematic diagram of PVA-RFP genome containing TuMV P3, NIa-Pro, or CP with 26 aa of CP and 18 aa of NIb at the 5′ and 3′ end of TuMV cistrons, respectively, for efficient cleavage of proteins [44]. (**B**) *Nicotiana benthamiana* leaves showing the expression of RFP from PVA-RFP with TuMV P3, NIa-Pro, or CP and GFP from TuMV-GFP. *N. benthamiana* leaves expressing PVA-RFP-TuMV-P3 or NIa-Pro abolished co-infection by TuMV-GFP, while PVA-RFP-TuMV CP efficiently permitted the co-infection by TuMV-GFP. Bars: 100 µm. (**C**) Stability assay of TuMV cistrons in the PVA genome. Agarose gel electrophoresis of RT-PCR products of TuMV cistrons from *N. benthamiana* leaves infiltrated with PVA-RFP harboring TuMV cistrons at 10 days post-agroinfiltration; lane 1. TuMV P3; lane 2. TuMV NIa-Pro; lane 3. TuMV CP; lane 4. PVA-RFP; lane 5: mock, and lane M: 1 kbp Plus DNA ladder (Invitrogen).

**Figure 6 viruses-15-01459-f006:**
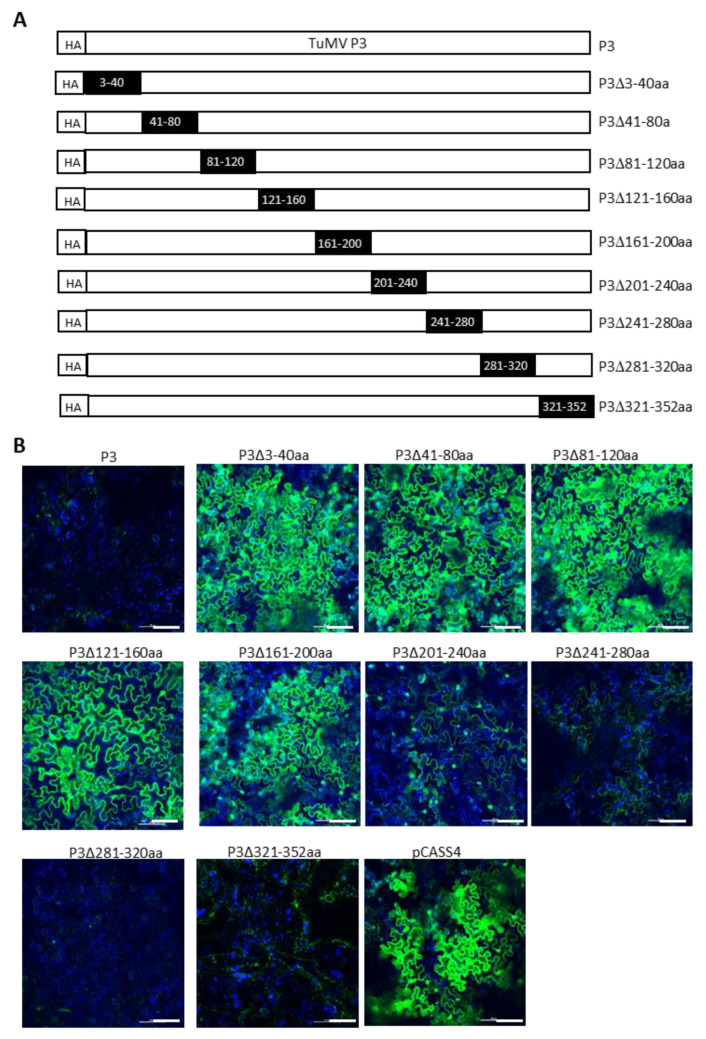
Turnip mosaic virus P3 amino acids 3 to 220 are required for SIE activity. (**A**) Diagrammatic representation of series of non-overlapping deletions in TuMV P3 cistron. Dark rectangle boxes indicate the location of amino acids deleted in the P3 cistron. (**B**) Superinfection exclusion assay of TuMV P3 deletion mutants. GFP expression in *Nicotiana benthamiana* leaves challenge-agroinfiltrated with pCB-TuMV-GFP into leaves pre-agroinfiltrated with pCASS4-TuMV P3 deletion mutants. Deletion of P3 amino acids 3 to 200 efficiently permitted infection by TuMV-GFP, while deletion of amino acids 201 to 240 showed a weak expression of GFP, depicting that some of the deleted amino acids in this region are required for SIE. Deletions comprising P3 amino acids 241 to 352 completely abolished superinfection by TuMV GFP, indicating that this region is dispensable for SIE activity. Bars: 100 µm.

**Figure 7 viruses-15-01459-f007:**
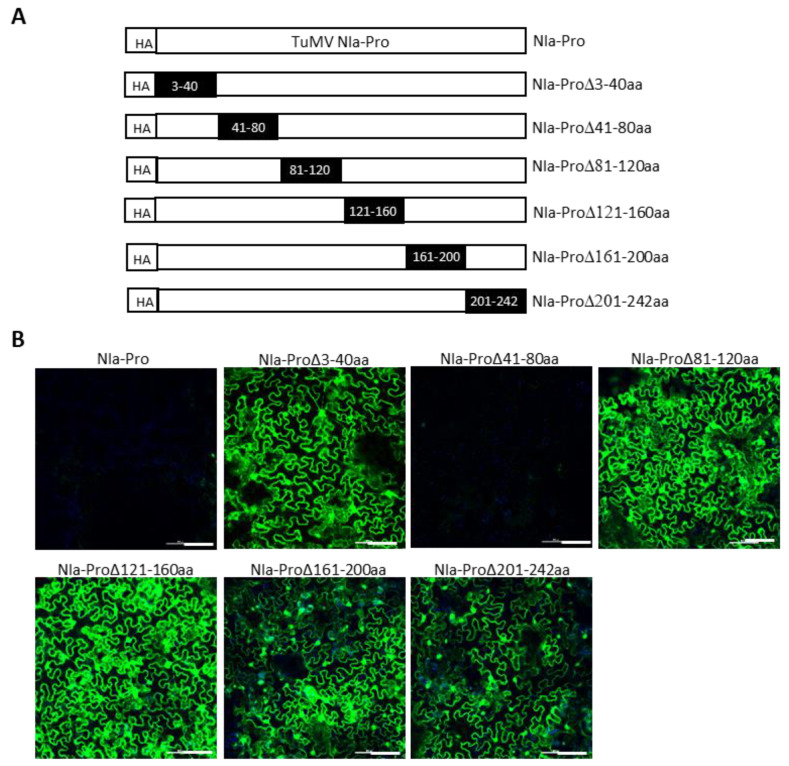
Mapping Turnip mosaic virus NIa-Pro amino acids required for SIE. (**A**) Schematic representation of a series of deletions in TuMV NIa-Pro. Dark rectangles represent deleted amino acids in the NIa-Pro cistron. (**B**) SIE assay of TuMV NIa-Pro deletion mutants. GFP expression in *Nicotiana benthamiana* leaves challenge-agroinfiltrated with pCB-TuMV-GFP into leaves pre-agroinfiltrated with NIa-Pro deletion mutants. TuMV NIa-Pro deletions comprising amino acids 3 to 40 and 81 to 242 did not elicit SIE, indicating that these amino acids are required for SIE. Bars: 100 µm.

## Data Availability

Not applicable.

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
