# Peer review of "P3 and NIa-Pro of Turnip Mosaic Virus Are Independent Elicitors of Superinfection Exclusion"

_viruses, 2023, doi:10.3390/v15071459_

Round 1

Reviewer 1 Report

Dear Authors,

I have an opportunity to review manuscript entitled “P3 and NIa-Pro of Turnip Mosaic Virus are Independent Elicitors of Superinfection Exclusion” submitted to the Viruses MDPI journal.

Authors concentrated on Superinfection exclusion, when an antagonistic interaction takes place between identical or closely related viruses in host cells. Authors tested individual TuMV-encoded proteins which were transiently expressed in the cells of Nicotiana benthamiana leaves, followed by challenged with a modified TuMV expressing the green fluorescent protein (TuMV-GFP).

Moreover, N. benthamiana cells expressing TuMV P3 or NIa-Pro did not express visible GFP fluorescence, suggesting that both of them could elicit potent SIE against TuMV-GFP. Furthermore, SIE elicitor activity of P3 and NIa-Pro were further confirmed by their heterologous expression from a different potyvirus, potato virus A (PVA).

The introduction part gives the reader sufficient background to analyse the obtained results, the hypothesis seems to be clear, but I warmly suggest to add precise aim of the studies at the end of introduction;

Materials and methods – is superinfection exclusion assay methodology described for the first time?

The results are clearly and exhaustively described, the microscopical documentation raises no objection and the quality is very good, but I warmly suggest  to add information that all microscopic visualization were conducted in epidermis leaves tissue; One question why results for TuMV were tested at 3dpi whereas for PVA at 10 dai ?

The discussion part is quite short and firm, I would like to ask about the future prospects coming form obtained results? And further steps in these interesting analysis;

Sincerely

Author Response

I have an opportunity to review manuscript entitled “P3 and NIa-Pro of Turnip Mosaic Virus are Independent Elicitors of Superinfection Exclusion” submitted to the Viruses MDPI journal.

Authors concentrated on Superinfection exclusion, when an antagonistic interaction takes place between identical or closely related viruses in host cells. Authors tested individual TuMV-encoded proteins which were transiently expressed in the cells of Nicotiana benthamiana leaves, followed by challenged with a modified TuMV expressing the green fluorescent protein (TuMV-GFP).

Moreover, N. benthamiana cells expressing TuMV P3 or NIa-Pro did not express visible GFP fluorescence, suggesting that both of them could elicit potent SIE against TuMV-GFP. Furthermore, SIE elicitor activity of P3 and NIa-Pro were further confirmed by their heterologous expression from a different potyvirus, potato virus A (PVA).

The introduction part gives the reader sufficient background to analyse the obtained results, the hypothesis seems to be clear, but I warmly suggest to add precise aim of the studies at the end of introduction; Materials and methods – is superinfection exclusion assay methodology described for the first time? The results are clearly and exhaustively described, the microscopical documentation raises no objection and the quality is very good, but I warmly suggest  to add information that all microscopic visualization were conducted in epidermis leaves tissue; One question why results for TuMV were tested at 3dpi whereas for PVA at 10 dai ?

Response: The aim of the present investigation was already added at the beginning of the last paragraph in the Introduction. The SIE method was explained in the materials and methods section and was not described for the first time for this virus. A sentence was added saying that all microscopic visualizations were conducted in epidermal cells of leaves (see lines 129-130).

Expression of GFP in TuMV-GFP challenge agroinfiltrated leaves was examined at 3 dai because GFP was observed in infiltrated leaves, but not in upper non-infiltrated leaves. In contrast, in the PVA system, the expression of GFP of TuMV-GFP was examined at 10 dai because we observed GFP in upper non-infiltrated systemic leaves.

The discussion part is quite short and firm, I would like to ask about the future prospects coming form obtained results? And further steps in these interesting analysis.

Response: Added a sentence as suggested. See lines 360-364.

Reviewer 2 Report

The authors used a transient overexpression assay to determine the SIE effector of a potyvirus TuMV. They find that P3 and NIa-Pro expression inhibits TuMV replication and that P3 and NIa-Pro are likely SIE effectors for TuMV. They further show that recombinant PVA-RFP carrying TuMV P3 and NIa-Pro confer SIE to TuMV-GFP. Some regions in P3 and NIa-Pro were later found responsible for SIE. This work demonstrates that TuMV encodes two spatially separated proteins that act independently to exert SIE on another superinfected TuMV. The manuscript is well-written, and the results are well-presented. Overall, this work is worthy of being published. However, some data need to be further explained, and minor figure labeling errors must be corrected before acceptance.

1.     TriMV and WSMV encode CP and NIa-Pro as SIE effectors. These two viruses belong to the same potyviridae family, the same as TuMV used in this work. The authors need to discuss whether the nature of different viruses or systems used in different studies determines the different outcomes of SIE effectors.

2.     In Figure 4, the authors observed locally expressed fluorescence of GFP and RFP to determine the non-occurrence of SIE between TuMV and PVA. However, this method can be complicated and interfered with by the transient GFP/RFP protein expression driven by the 35S promoter. To get a firm conclusion that TuMV and PVA do not induce mutual SIE, the authors are suggested to provide images of systemically infected leaves from plants co-infected with TuMV-GFP and PVA-RFP with a wide / zoomed-out field of view under the scope.

3.     In the lower panels of Figure 5B, the label should be PVA-RFP-TuMV CP according to the figure legend and main text.

4.     In the study shown in Figure 5, when PVA carries TuMV P3, the SIE observed by analyzing RFP and GFP signals seems incomplete, especially compared with the figure when using full genomic TuMV to exert the SIE (Fig. 1B). Can the authors explain the different level of SIE between overexpression assay and recombinant PVA expression assay?

5.     In panel A from Fig.6 and Fig. 7, the expression of P3 or NIa-Pro mutants should be presented by detecting HA-tag using western blot. Similarly, the GFP accumulation level in panel B of Fig. 6 and Fig. 7 is suggested to be shown in western blot.

Author Response

The authors used a transient overexpression assay to determine the SIE effector of a potyvirus TuMV. They find that P3 and NIa-Pro expression inhibits TuMV replication and that P3 and NIa-Pro are likely SIE effectors for TuMV. They further show that recombinant PVA-RFP carrying TuMV P3 and NIa-Pro confer SIE to TuMV-GFP. Some regions in P3 and NIa-Pro were later found responsible for SIE. This work demonstrates that TuMV encodes two spatially separated proteins that act independently to exert SIE on another superinfected TuMV. The manuscript is well-written, and the results are well-presented. Overall, this work is worthy of being published. However, some data need to be further explained, and minor figure labeling errors must be corrected before acceptance.

  1. TriMV and WSMV encode CP and NIa-Pro as SIE effectors. These two viruses belong to the same potyviridae family, the same as TuMV used in this work. The authors need to discuss whether the nature of different viruses or systems used in different studies determines the different outcomes of SIE effectors.

Response: Added a sentence as suggested. See lines 332-334.

  1. In Figure 4, the authors observed locally expressed fluorescence of GFP and RFP to determine the non-occurrence of SIE between TuMV and PVA. However, this method can be complicated and interfered with by the transient GFP/RFP protein expression driven by the 35S promoter. To get a firm conclusion that TuMV and PVA do not induce mutual SIE, the authors are suggested to provide images of systemically infected leaves from plants co-infected with TuMV-GFP and PVA-RFP with a wide / zoomed-out field of view under the scope.

Response: We agree with the reviewer. We already examined GFP and RFP from systemic leaves. This was mentioned in the text in the original manuscript (see lines 229-231). Figure 4 legend was modified to reflect the reviewer’s suggestion, and this was updated in the revised manuscript. Please see the Figure 4 legend.

  1. In the lower panels of Figure 5B, the label should be PVA-RFP-TuMV CP according to the figure legend and main text.

Response: Thank you for catching this. The label was PVA-RFP-TuMV-CP. But it was not visible. In the revised figure, we expanded the label box to visible PVA-RFP-TuMV-CP. See figure 5.

  1. In the study shown in Figure 5, when PVA carries TuMV P3, the SIE observed by analyzing RFP and GFP signals seems incomplete, especially compared with the figure when using full genomic TuMV to exert the SIE (Fig. 1B). Can the authors explain the different level of SIE between overexpression assay and recombinant PVA expression assay?

Response: We added a sentence this discrepancy might be due to the stability of inserts and/or low-level expression through PVA. See lines 278-280. 

  1. In panel A from Fig.6 and Fig. 7, the expression of P3 or NIa-Pro mutants should be presented by detecting HA-tag using western blot. Similarly, the GFP accumulation level in panel B of Fig. 6 and Fig. 7 is suggested to be shown in western blot.

Response: We already showed Western blots with wild-type P3 and NIa-Pro cistrons for the expression of these proteins and the lack of detection of TuMV with GFP antibody. We respectfully disagree with the reviewers’ suggestion to show the Western blots with HA and GFP antibodies. These experiments may not provide different or additional results.